# Nerve Density and Neuronal Biomarkers in Cancer

**DOI:** 10.3390/cancers14194817

**Published:** 2022-10-01

**Authors:** Shahrukh R. Ali, Madeleine Jordan, Priyadharsini Nagarajan, Moran Amit

**Affiliations:** 1The University of Texas Medical Branch, Galveston, TX 77555, USA; 2Head and Neck Surgery Department, The University of Texas MD Anderson Cancer Center, Houston, TX 77030, USA; 3Pathology Department, The University of Texas MD Anderson Cancer Center, Houston, TX 77030, USA

**Keywords:** neoplasms, cancer, tumor microenvironment, nerve tissue, nerve pathology, nerve density, nerves, neural factors

## Abstract

**Simple Summary:**

Researchers have shown that tumor biomarkers and increased nerve density are important clinical tools for determining cancer prognosis and developing effective treatments. The aims of our review were to synthesize these findings by detailing the histology of peripheral nerves, discuss the use of various neuronal biomarkers in cancer, and assess the impact of increased nerve density on tumorigenesis. This review demonstrates that specific neuronal markers may have an important role in tumorigenesis and may serve as diagnostic and prognostic factors for various cancers. Moreover, increased nerve density may be associated with worse prognosis in different cancers, and cancer therapies that decrease nerve density may offer benefit to patients.

**Abstract:**

Certain histologic characteristics of neurons, novel neuronal biomarkers, and nerve density are emerging as important diagnostic and prognostic tools in several cancers. The tumor microenvironment has long been known to promote tumor development via promoting angiogenesis and cellular proliferation, but new evidence has shown that neural proliferation and invasion in the tumor microenvironment may also enable tumor growth. Specific neuronal components in peripheral nerves and their localization in certain tumor sites have been identified and associated with tumor aggressiveness. In addition, dense neural innervation has been shown to promote tumorigenesis. In this review, we will summarize the histological components of a nerve, explore the neuronal biomarkers found in tumor sites, and discuss clinical correlates between tumor neurobiology and patient prognosis.

## 1. Introduction

The tumor microenvironment consists of well-recognized interactions between tumor cells and the surrounding vasculature, immune system, stromal cells, and extracellular matrix [1]. Characterization of these interactions has allowed clinical researchers to develop numerous oncologic therapeutic agents ranging from vascular endothelial growth factor inhibitors to immunotherapy [2]. The interaction between the nervous system and the tumor microenvironment has not received as much attention in the literature but has recently garnered interest [3]. Studies have measured the neural density in the tumor microenvironment and elucidated its association with tumorigenesis. Moreover, researchers have also begun to identify specific neural biomarkers in these dense neural networks [4]. These biomarkers may allow us to trace the origins of a tumor, differentiate closely related cancers, and predict the prognosis of a patient. In this review, we will discuss the histology of peripheral nerves, focus on specific biomarkers located in extracranial tumors, and summarize clinical correlates between tumor neurobiology and patient prognosis.

## 2. Histology of Peripheral Nerves

Macroscopically, healthy peripheral nerves appear as white fascicular bundles surrounded by pale, thin connective tissue; the thickness or diameter typically decreases progressively towards the distal aspect [5]. However, previously traumatized or damaged nerves appear thinner, with a prominent grayish hue [6]. Histologically, a peripheral nerve is composed of neuronal axons concentrically surrounded by endoneurium and perineurium and bundled together by epineurium, as illustrated in Figure 1 [7]. In a cross-section of a typical peripheral nerve, Schwann cells comprise the majority of the visible nuclei, followed by fibroblasts, mast cells, and, rarely, endothelial cells.

Each nerve fiber is composed of a central axon, a cylindrical extension of cytoplasm or perikaryon of the neural body located in the ganglia. The central axon is enveloped by Schwann cells along its entire length [7]. The majority of the axon consists of a delicate endoskeleton composed of a longitudinally oriented lattice of actin, neurofilaments, and tubulin (Figure 2), which enmesh cellular organelles such as mitochondria, smooth endoplasmic reticulum, peroxisomes, rare ribosomes, and neurotransmitter vesicles [8]. Neurofilaments are classified into three major subunits based on their molecular weights: neurofilament-light, neurofilament-medium, and neurofilament-heavy (Figure 2) heteropolymers [9,10]. Therefore, immunostaining for neurofilaments may be used to identify nerve fibers and neural proliferation, specifically neurite branching. However, smaller unmyelinated nerve fibers may be difficult to identify owing to the paucity of axons and their size; therefore, markers of other nonneural cells are frequently used for the identification of nerves.

In myelinated nerve fibers, a single Schwann cell concentrically encircles repeating segments of the neuronal axon. Each Schwann cell produces a myelin sheath encapsulating a single segment of neuronal axon [7,11]. Unmyelinated gaps on the axon, known as Nodes of Ranvier, are essential for saltatory (discontinuous) and very rapid conduction of depolarization through the nerve fiber [12,13]. In nonmyelinated nerves, a single Schwann cell may encircle several axons. Because Schwann cells are the most numerous nucleated cells in a peripheral nerve, their presence has been frequently used for the identification of nerves in histologic sections. Schwann cells are often positive for S100 (nuclear and cytoplasmic pattern) (Figure 3), GFAP, SOX10, E-cadherin, CD56, CD57, PGP9.5, and calretinin [14,15].

Endoneurium encloses axons and Schwann cells, as well as stromal components including collagen fibers, fibroblasts, and CD34-positive dendritic cells [7,16]. The endoneurial collagen is organized as an outer sheath of larger, longitudinally oriented collagen fibers and an inner sheath of fine oblique and circumferential collagen fibers. The inner sheath and Schwann cell basement membrane fuse to form the neurilemma.

Perineurium is a tubular sheath encircling several endoneurial components to form nerve fascicles and is composed of flattened cells and collagen [7,17]. The perineurial cells are characterized by numerous tight junctions, thereby producing a blood-nerve barrier. Immunostaining for markers of perineurial cells such as claudin-1, mucin-1 (MUC1, also known as epithelial membrane antigen [EMA]), and glucose transporter protein-1 (GLUT-1) is also commonly used in the identification of peripheral nerves as well as neoplasms originating from perineurium [17].

Epineurium is composed of connective tissue and binds several nerve fascicles together, sometimes along with small blood vessels and lymphatics, to form a neurovascular bundle [18]. It is typically composed of collagen and a few elastin fibers [18]. In larger nerves, the blood vessels feeding nerves, i.e., the vasa nervosum, run longitudinally within the epineurium. The epineurium is thicker proximally; however, in the distal aspect, where only single nerve fascicles are present, the epineurium may disappear completely.

## 3. Perineurial Invasion

Although involvement of nerves has been well recognized as a mode of cancer spread for almost 2 centuries [19,20], the criteria for histopathologic identification of perineurial invasion (PNI), a subset of neurotropism, continue to be debated [21,22]. Currently, PNI (Figure 4) is defined as the presence of tumor cells abutting or in close proximity to a nerve with encirclement of at least a third of the nerve circumference by tumor or the presence of cancer cells within the epineurial, perineurial, and/or endoneurial compartments of a nerve [23]. However, some pathologists consider mere abutting of tumor cells against a nerve—without intervening stroma—to constitute PNI, particularly in melanoma [24], while others require the unequivocal presence of tumor cells within the perineurium [21]. Additionally, caution should be employed during radiologic [25] and histopathologic [26] evaluation because some benign entities may show features similar to PNI.

PNI has been demonstrated to be an independent risk factor for local tumor recurrence and, in some contexts, shorter patient survival [23,27]. In addition, its presence has been associated with poor prognosis in cancers of the pancreas [28,29] including the ampulla [30]; gallbladder [31]; extrahepatic bile ducts [32]; stomach [33]; colon and rectum [34,35]; penis [36]; and prostate [37]; in melanoma [38,39,40]; in cancers of several head and neck sites, including the larynx [41], oral cavity [41], salivary glands [41,42], and pharynx [41]; and in cutaneous squamous cell carcinoma [43]. Head and neck cancers also have a high propensity for involving cranial nerves and for intracranial spread, resulting not only in functional and cosmetic morbidities but also carrying a high risk for tumor-specific mortality [42,44,45,46]. Moreover, the presence of PNI outside the confines of a main tumor mass (extratumoral PNI) may be associated with worse prognosis [47].

Invasion of small unnamed branches of cranial nerves or most peripheral nerves may not be clinically symptomatic, in particular when the nerve fiber diameter is less than 0.1 mm [48]. This size criterion applies primarily to cutaneous squamous cell carcinoma of the head and neck, where PNI in the subcutis is also considered to be an indicator of poor prognosis [48,49] and is included as a criterion for T categorization in the eighth edition of the AJCC staging system [50]. Similarly, PNI is also included in the T categorization of penile cancers [51]. Therefore, PNI of any size should be reported when identified. Additionally, it is important to document when there are multiple foci of PNI (Figure 4).

## 4. Determination of Nerve Density

Nerve density is defined as the number of nerve profiles identified within a specific area and is expressed as the number per square millimeter or per high-power field (200× or 400×) [52]. Higher nerve density, in general, appears to correlate with worse prognosis. In most studies published thus far, at least one immunohistochemical marker has been used for identification of nerves [52]. While use of immunohistochemical markers increases the sensitivity for identifying nerves, individual immunostaining may not be widely available. Therefore, we propose a standardized method to accurately determine nerve density by hematoxylin and eosin (H&E) stains using scanned whole-slide images (WSI) (Figure 5). In the example illustrated here, a representative H&E-stained section including tumor as well as surrounding uninvolved tissue was scanned using a ScanScope digital pathology system (Aperio, Vista, CA, USA) in SVS format at a minimum magnification of 20× and visualized using ImageScope viewer (Aperio). All nerve profiles were annotated (green arrows) with specific notations for PNI (turquoise rectangles). The extent of viable tumor (yellow line), excluding nonviable tissue such as anucleate keratinous material (asterisks) and uninvolved tissue (red line), was drawn on the WSI. The respective tumor and normal areas were determined from the annotations, and the nerve density in each slide was calculated as the total number of nerves divided by area and expressed as the number per square millimeter.

This standardized way of determining nerve density, however, has two limitations: (i) facilities for WSI are not yet widely available, and (ii) small nerve fascicles may not be discernible on H&E-stained sections. Moreover, it is currently unclear whether evaluation of a single representative section would be sufficient and/or scientifically valid, and the criteria for selecting the representative section are subjective.

## 5. Mechanism of Interaction between Nerves and Cancer

Nerves play an important role in the microenvironment of tumors. In recent studies, PNI has been shown to be associated with tumor progression and poor outcomes. The mechanism of tumor-nerve interaction is postulated to involve two mechanisms. Tumor cells secrete neurotrophic factors, neurotransmitters, and axon guidance molecule, signaling initiation of neuron reprogramming, which increases nerve recruitment or infiltrates previously existing nerves. In addition, nerves also secrete neuroactive molecules that interact with receptors on tumor cells and on cells in the tumor microenvironment. These interactions have been shown to promote tumorigenesis. 

### 5.1. Effect of Tumor Cells on Nerves 

Cancer cells have been shown to infiltrate into or around nerves during PNI. Cancer cells can use PNI as a pathway to metastasize, similar to vascular and lymphatic channels [53]. More specifically, the recent literature has defined the neural tracking hypothesis as the likely process of PNI. Tumor cells track along a nerve or nerve fiber and inflict nerve injury after infiltrating the perineural space. The damage to the perineurium causes the release of multiple inflammatory cytokines, creating a unique cellular and biochemical microenvironment known as the perineural niche. This niche further promotes neural regeneration, which can act as a pathway for metastasis. Moreover, cancer cells can react to neurogenesis and increase the recruitment of axons into tumor tissue. The recruitment of nerves into these tumor sites, such as angiogenesis, increases tumorigenesis. These newly recruited nerves have both direct and indirect effects on tumor sites. In terms of direct effects, neurons can release neurotransmitters directly into a synapse with tumor cells, which can propagate tumor progression. In the paracrine mode, neurotrophic factors such as nerve growth factor (NGF), brain-derived growth factor (BDNF), glial cell line-derived neurotrophic factors, axon guidance molecules such as CCL2, CX3CL1, EphA2m, and Slit, and neurotransmitters including SCh, glutamate, glycine, epinephrine, norepinephrine, and dopamine are released via nerves [54,55]. Tumor cells express receptors such as tyrosine kinase receptor A (TrkA), TrkB, and NGF receptor (NGFR) that interact with these factors, leading to a downstream cascade that increases tumor proliferation. In addition to the paracrine mode, the chemical synapse is a unique form of interaction between nerves and tumors. In these synapses, two adjacent neurons communicate using transmitters such as glutamate. Glutamate, which functions as an excitatory neurotransmitter, can cause the depolarization of neurons. This depolarization in nerves has been shown to increase tumor proliferation [53]. 

### 5.2. Indirect Effects of the Nerves on Tumors

Nerves also have indirect effects on tumors. They have been shown to interact with stromal components in the tumor microenvironment, indirectly promoting tumor growth and metastasis. For example, nerves can secrete neurotransmitters such as catecholamine, Ach, dopamine, NGF, and BNF, which all promote angiogenesis [56,57,58,59,60]. This angiogenesis can provide an essential vascular supply to tumor sites, allowing for rapid cell proliferation. In addition, nerves communicate with the immune system and can contribute to tumor progression. For example, in the spleen, adrenergic innervation has been shown to stimulate the production of Ach in T-cells [61]. Ach secreted from T-cells can inhibit tumor necrosis factor production, thereby increasing tumor growth. In addition, in the lungs, T-cell-derived Ach has been shown to accelerate tumor growth and metastasis [62].

## 6. Neural Biomarkers and Their Role in Cancer

Neural biomarkers can be used to identify the origin of a tumor, can enable the differentiation and classification of tumors, and can serve as prognostic factors in several cancers [63]. These biomarkers are most often found in immature neurons, which do not have a fully developed initial axon segment, unlike mature neurons, which do have a developed axon segment. Moreover, immature neurons are undifferentiated, whereas mature neurons have evolved into specialized cells with specific tasks. In this section, we will further explore these neural biomarkers and their involvement in cancer.

### 6.1. Biomarkers in Immature Neurons

#### 6.1.1. NeuroD1

Neurogenic differentiation factor 1 (NeuroD1) is a basic helix–loop–helix transcription factor found in neurons that plays an important role in neuronal differentiation [64]. NeuroD1 has been shown to play an important role in the tumorigenesis of various peripheral tumors such as schwannomas and neuroblastomas.

In a study assessing the impact of NeuroD1 in neuroblastoma, researchers examined the expression profile of NeuroD1 in MYCN-overexpressing transgenic mice and the role of NeuroD1 in tumor formation [65]. They found that NeuroD1 was strongly expressed in the celiac sympathetic ganglion of MYCN mice and expressed in all subsequently generated advanced cancer tissue. Moreover, the authors noted that inhibition of NeuroD1 via short hairpin RNA (shRNA) resulted in decreased neuroblastoma cellular motility. They also noted that tumor proliferation was suppressed with NeuroD1 inhibition. The researchers concluded that NeuroD1 expression is associated with increased tumorigenesis of neuroblastoma and associated with a poor prognosis in mouse models [65].

Investigators in another study used a variety of transgenic mouse lines to determine how the expression of NeuroD1 affects schwannoma tumor progression, vestibular function, and schwannoma cell proliferation [66]. The authors found that gene transfer of *NEUROD1* significantly reduced the proliferation of schwannoma cells [66]. The researchers then deleted the neurofibromatosis 2 suppressor gene (*NF2*) in schwannoma cells and, as expected, noted increased intraganglionic schwannoma cell proliferation [66]. Interestingly, in these intraganglionic schwannoma cells, addition of NeuroD1 induced variable effects, with no overt decrease in schwannoma cell proliferation [66]. The researchers next performed sciatic nerve axotomy to determine the effect of Neuro D1 on peripheral nerves [66]. Axotomy significantly increased schwannoma cell proliferation, as expected [66]. The authors concluded that NeuroD1 inhibits the proliferation of Schwann cells and functions as a tumor suppressor. However, they noted that the results were mixed and that further research is needed to explore NeuroD1′s potential role as a therapeutic agent [66].

#### 6.1.2. Tubulin Beta-3 Chain

The protein tubulin beta-3 chain (TUBB3) has many critical cellular functions, including roles in structural support, protein delivery, and cell division [67]. The expression of TUBB3 in cancers has also been postulated to play a role in resistance to taxane-based chemotherapy and thus is of great interest to researchers [67]. In one study, immunohistochemical expression of TUBB3 was assessed in 3911 tissue samples from 100 different tumor categories and 76 different normal tissue types [67]. The study found that all neuroblastoma samples strongly expressed TUBB3. Thus, the investigators suggested that clinicians should consider prescribing taxane-based therapy in patients with neuroblastoma.

In another study, researchers investigated the impact of TUBB3 expression in neurogenic cancers [60]. They noted differing patterns of expression of the TUBB3 protein in the various cancers. In both embryonic and adult neuronal tumors of the peripheral nervous systems, TUBB3 expression was associated with lower rates of neuronal proliferation caused by higher rates of neuronal differentiation. In contrast, in non-neurogenic tumors such as lung cancer, the presence of TUBB3 increased the histologic grade of tumors and promoted cellular proliferation. Thus, in neurogenic tumors, TUBB3 may act as a protective factor by promoting cellular differentiation and thereby attenuating tumorigenesis [68].

#### 6.1.3. Stathmin 1

Stathmin 1 (STMN1) belongs to a family of proteins that are important regulators of microtubule dynamics [69]. Stathmin 1 depolymerizes and prevents the polymerization of several different microtubules. Recently, researchers have begun to take an interest in this protein and have noted its involvement in neurogenic cancers [69].

A study found that stathmin 1 suppression reduced neuroblastoma cell invasion into the extracellular matrix and that its role in tumor invasion is mediated by RHO-associated protein kinase (ROCK), a key regulator of cell migration [70]. In neuroblastoma cells, the suppression of ROCK inhibited cell migration. Moreover, reduced stathmin 1 expression in neuroblastoma cells significantly increased the activity of transforming protein RhoA, which is upstream of ROCK, and induced expression of the RhoA/ROCK pathway. Finally, stathmin 1 suppression in neuroblastoma tumor models decreased whole-body metastasis in the lung, kidney, and liver. Thus, the role of stathmin 1 in neuroblastoma tumorigenesis makes it a potential target for cancer therapeutics [70].

### 6.2. Biomarkers in Mature Neurons

#### 6.2.1. Microtubule-Associated Protein 2

Microtubule-associated protein 2 (MAP2) belongs to the family of microtubule-associated proteins, which are thought to be involved in assembling microtubules and to play a crucial role in neurogenesis [71]. MAP2 proteins have also been shown to be expressed in neuronally differentiated neurons [72] A study seeking to establish the neural features of peripheral neuroblastic tumors analyzed samples of 12 neuroblastomas, 2 ganglioneuroblastomas, and 4 ganglioneuromas. All tumor samples expressed MAP2, suggesting that MAP2 may be a biomarker for identifying tumors with neural origins or those with significant neural involvement [73].

Another study assessed whether MAP2 expression could be used in the diagnosis of neuroblastoma [74]. Researchers used immunohistochemical analyses of tissue microarrays to evaluate the utility of a commercially available antibody against MAP2 in detecting primary and metastatic neuroblastomas. They found that MAP2 showed cytoplasmic reactivity in 95% of primary and 100% of metastatic neuroblastomas. In contrast, MAP2 was not found in other small round blue cell tumors. Additionally, in normal tissue, MAP2 was expressed only in organs of neural crest origin such as the adrenal medulla. Thus, the authors concluded that MAP2 is both a sensitive and specific marker to detect neuroblastoma and can differentiate neuroblastoma from other tumors with similar morphological features [74].

#### 6.2.2. Synaptophysin

In a study evaluating the utility of synaptophysin for detecting childhood neuroblastoma, synaptophysin immunoreactivity was found in six of six neuroblastoma samples tested but not in other small round cell tumors with similar features [75]. In addition, rhabdomyosarcomas, lymphomas, and Ewing sarcomas were all negative for synaptophysin. This study indicated that synaptophysin could be an important diagnostic marker for evaluating neuroblastoma and should be included in marker panels [75].

## 7. Clinical Evidence of Nerve Density in Cancer

Neural density has been shown in the literature to be associated with increased tumor aggressiveness and proliferation [52]. In this section, we focus on clinical studies that have evaluated the clinical relevance of neural density in peripheral cancers. One such cancer in which researchers have noted increases in neural density is prostate cancer. A retrospective analysis of 43 patients with prostate cancer showed that a high nerve density index was associated with a higher tumor proliferation index [76]. Another study was conducted to identify the spatial and temporal associations of nerve density with preneoplastic and neoplastic growth of the prostate [77]. These researchers found that nerve density was higher in the neoplastic and preneoplastic regions than in normal prostate tissues [77]. They further showed that cancer cells interacted with neurons and promoted the growth of neurites [77]. The study suggested that neurogenesis is a key factor contributing to aggressiveness and recurrence in prostate cancer [77].

Nerve density has also been postulated to play a role in head and neck cancer [3]. A study that analyzed retrospective samples from oral cancers revealed that tumors that lacked TP53 expression displayed increased neurogenesis [3]. The authors noted that a loss of TP53 caused transdifferentiation of peripheral sensory nerves, which initiated adrenergic neurogenesis. The subsequent increase in neural density was associated with poor clinical outcomes, suggesting that neurogenesis may be an important target for cancer therapeutics in the future [3].

Another study investigated the innervation of thyroid cancers and the association of neural density with outcomes in patients [78]. Researchers detected nerves in papillary thyroid cancer by immunohistochemistry using the pan-neuronal marker PGP9.5 and compared nerve densities to those found in papillary and follicular thyroid carcinomas and benign thyroid tissue. Interestingly, nerves were present in both benign thyroid tissue and cancer, but nerve density was significantly higher in papillary thyroid carcinoma. However, nerve density was not higher in follicular thyroid carcinoma and benign thyroid tissue. The researchers also noted that neural density was significantly associated with extrathyroidal invasion, suggesting that high neural density may be associated with worse clinical outcomes [78].

Breast cancer has also been shown to exhibit high nerve density [79]. Researchers in one study evaluated a total of 196 samples including 20 normal tissue specimens, 14 fibroadenomas, 18 ductal carcinomas in situ (DCIS), and 144 invasive ductal carcinomas. Immunostaining for PGP 9.5 and S100 to identify nerves showed PGP9.5-positive nerve fibers in all normal tissue samples and 61% of invasive ductal carcinomas. However, PGP9.5 staining was not found in fibroadenomas or DCIS. Moreover, patients with less than 3 years disease-free survival tended to have higher expression of PGP9.5 than did patients with 3 or more years of disease-free survival, suggesting that dense innervation may be associated with breast cancer progression [79].

In another study, researchers evaluated whether nerve fiber density could differentiate between schwannomatosis and neurofibromatosis [80]. Patients with schwannomatosis have more neuropathic pain than do patients with neurofibromatosis, but the presence of pain is the only clinical differentiator between the two diagnoses. Thus, the researchers examined skin biopsy specimens from 34 patients with schwannomatosis and 25 patients with neurofibromatosis type 2. In the schwannomatosis group, 97% of patients showed intraepidermal nerve fiber density (IEND) below or at levels similar to a matched reference group (*p* < 0.0001). In contrast, in patients with neurofibromatosis type 2, only 44% had lower IEND compared to the corresponding reference group. Thus, quantifying nerve density is another clinical tool with which to differentiate schwannomatosis from neurofibromatosis and to develop therapeutics that can treat the neuropathic pain associated with schwannomatosis [80].

In another study conducted to develop a new parameter to evaluate peripheral nerve pathology on ultrasound [81], researchers developed software to quantify the ratio between hyperechoic and hypoechoic regions on ultrasound. Nerve density was defined as the ratio of hypoechoic pixels to total pixels. The study included 65 patients, 35 of whom had been diagnosed with carpal tunnel syndrome and 30 with neurofibromas. The study showed that ultrasound could detect differences in nerve density among normal nerves, nerves affected by carpal tunnel syndrome, and neurofibromas, and the results were statistically significant [81]. This study thus discovered a novel method to differentiate nerve density and offered an alternative diagnostic tool that can differentiate neural pathologies.

Finally, another study evaluated the effect of vasoinhibin on NGF-induced differentiation and survival of PC12 pheochromocytoma cells [82]. Vasoinhibin has been shown to decrease NGF-induced outgrowth in peripheral neurons, but there is limited research on the effect of vasoinhibin on developing neurons. The authors therefore evaluated whether administration of recombinant vasoinhibin or lentiviral-transduced vasoinhibin affected neural differentiation and growth. They found that vasoinhibin significantly reduced NGF-induced neurite outgrowth and cell survival and increased apoptosis. The authors concluded that therapeutic agents that decrease neural density may be effective and that future clinical trials should further evaluate these findings [82].

## 8. Discussion

The roles of nerve histology, neural biomarkers, and increased nerve density in tumor development have been the subject of many recent studies with aspirations of expanding the repertoire of cancer therapies. Recent evidence supports the involvement of these factors in tumorigenesis, tumor proliferation, angiogenesis, and tumor invasion.

This review described the histological components of a peripheral nerve: nerve fibers concentrically surrounded by endoneurium, and perineurium bundled together by epineurium. Immunostaining for histologic components of perineurial cells and epineural cells has been commonly used in the identification of peripheral nerves and neoplasms that may originate from perineurial cells [17].

This review also considered research demonstrating that specific neuronal markers may have an important role in tumorigenesis and can serve as important diagnostic and prognostic factors for extracranial cancers. These markers are categorized based on whether they originate from immature or mature neurons [63].

We also assessed recent clinical evidence of neural density and its role in tumorigenesis. In many cancers, such as certain prostate, oral, thyroid, and breast cancers, high neural density is associated with tumor invasion and aggressiveness and with poor clinical outcomes [79]. We found that neural density may also serve as an important differentiator between closely related pathologies such schwannomatosis and neurofibromatosis or neurofibromas and carpal tunnel syndrome. We also noted that agents that decrease neural density such as vasohibin can attenuate proliferation and increase apoptosis of pheochromocytoma cells, thus serving as an effective oncologic therapy [82]. These results indicate that tumorigenesis is associated with increased neural density which could be targeted with future pharmacotherapeutic agents.

We encourage future research to further develop methods to identify the histological characteristics of nerves, investigate additional neuronal biomarkers, and develop clinical trials that assess oncologic therapies targeting neural density. The current method to detail the characteristics of nerves involves semi-quantitative analysis, which has several limitations [82]. These limitations range from variability in methods between researchers in performing the staining to different interpretations of the stains^.^ [82]. Standardization and guidelines are needed to ensure that staining results are precise, accurate, and consistent [82]. In addition, the literature on identifying tumor biomarkers in peripheral neurogenic tumors is scarce in comparison to that on tumors with different tissues of origin. Thus, researchers may find benefit in investing resources in delineating new biomarkers for peripheral neurogenic tumors. Lastly, increased nerve density has been shown to worsen prognosis for peripheral cancers in preclinical models, but there are currently no clinical studies assessing this effect. Thus, we encourage researchers to develop clinical trials that assess the impact of increased nerve density in addition to oncologic therapeutics that decrease nerve densities.

## 9. Conclusions

This review has detailed the histology of peripheral nerves, discussed the use of various neuronal biomarkers in extracranial tumors, and assessed the impact of increased nerve density on tumorigenesis in these tumors. We reviewed research showing biomarkers with potential to serve as diagnostic and prognostic tools in neurogenic tumors of the peripheral nervous system. In addition, we noted evidence demonstrating that increased neural density may be associated with tumorigenesis and that treatments that reduce neural density may show potential in treating peripheral tumors. Future studies should further develop clinical trials that assess the impact of targeting neural density in peripheral tumors and assess whether these strategies can improve patients’ prognosis.

## Figures and Tables

**Figure 1 cancers-14-04817-f001:**
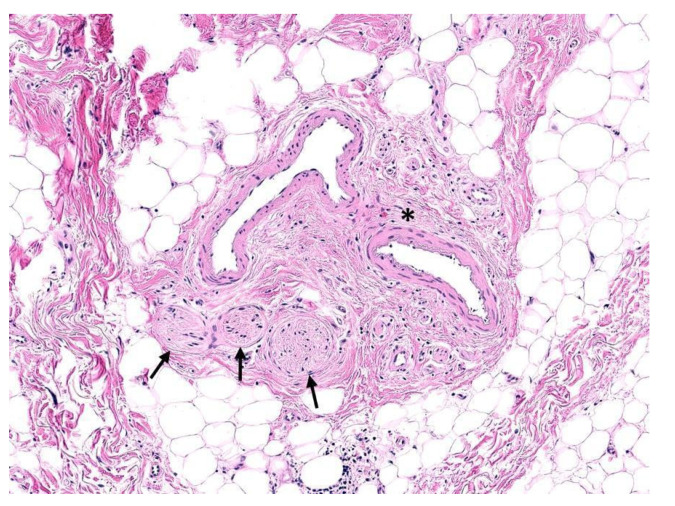
Neurovascular bundle. Several nerve fibers of varying sizes, each surrounded by a thin layer of perineurium (arrows) in association with blood vessels ensheathed in epineurium (asterisk) within fibroadipose tissue (hematoxylin and eosin, 200×).

**Figure 2 cancers-14-04817-f002:**
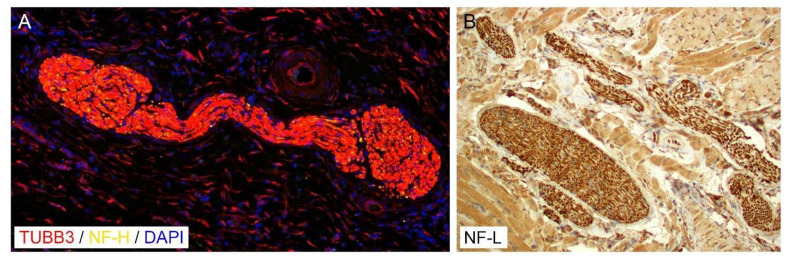
Expression of (**A**). Beta III tubulin (TUBB3) and neurofilament-heavy (NF-H) in a subcutaneous nerve, counterstained with DAPI (immunofluorescence, 400×); (**B**). Neurofilament-light (NF-L) in nerve fibers within tongue musculature, counterstained with hematoxylin (immunohistochemistry, 200×).

**Figure 3 cancers-14-04817-f003:**
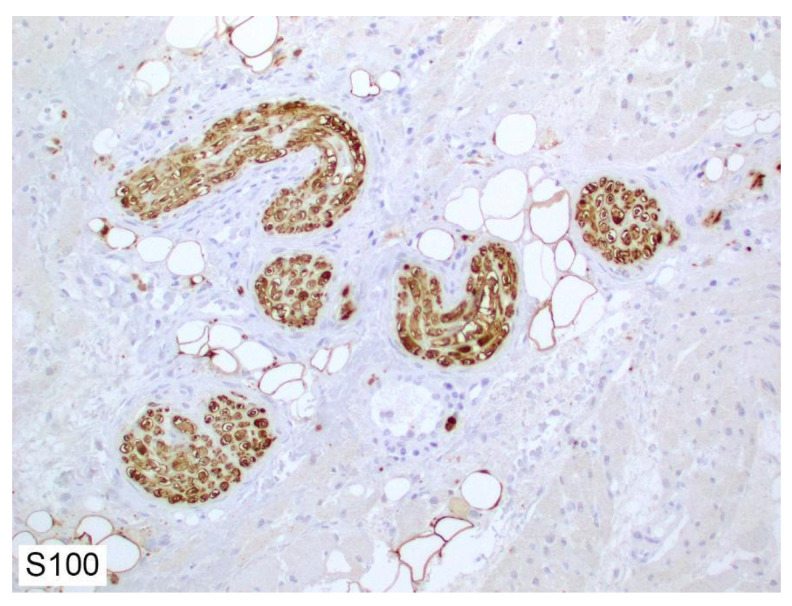
S100 protein expression in nerves located within superficial facial muscles, counterstained with hematoxylin (immunohistochemistry, 200×).

**Figure 4 cancers-14-04817-f004:**
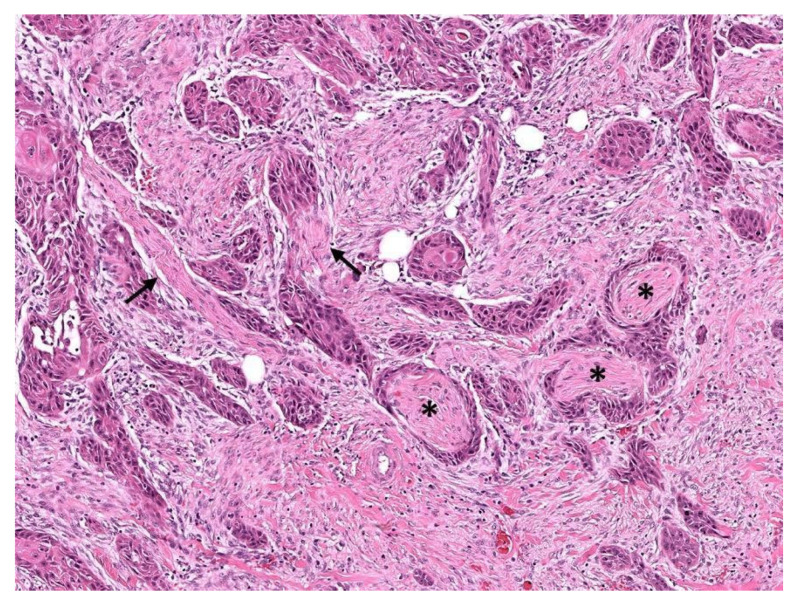
Multifocal perineurial invasion, involving small (arrows, 0.03 mm) and medium (asterisks, 0.12 mm) caliber nerve fibers in cutaneous squamous cell carcinoma of the right temple (hematoxylin and eosin, 100×).

**Figure 5 cancers-14-04817-f005:**
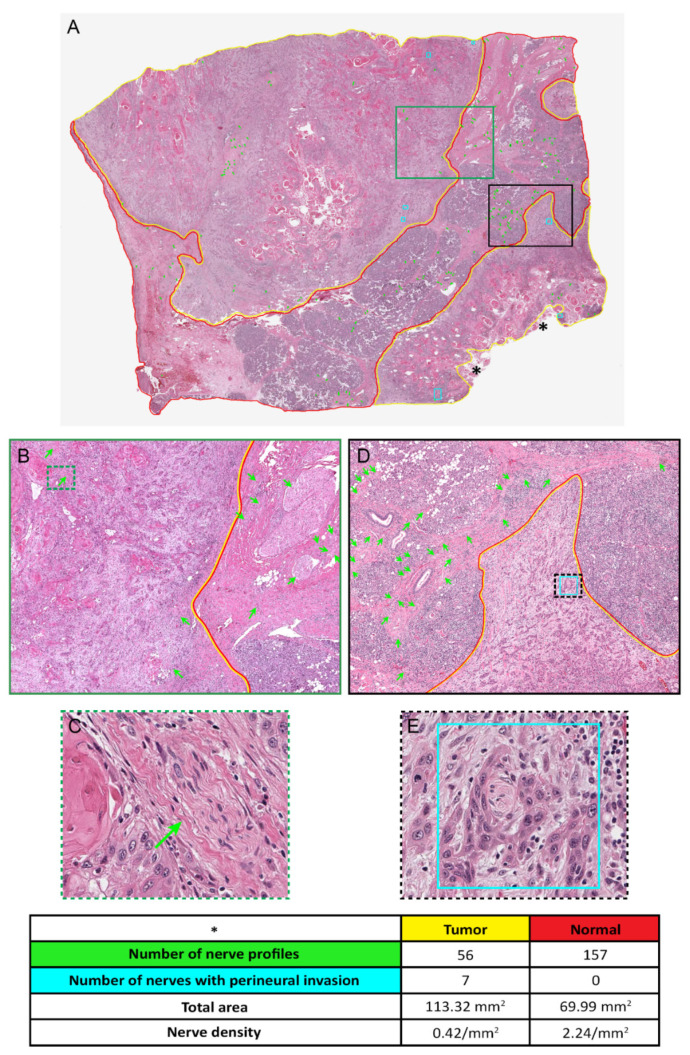
Determination of nerve density. (**A**). Whole-slide image of representative section of squamous cell carcinoma metastatic to parotid gland, scanned at 20× magnification. The areas of tumor (yellow line) excluding anucleate keratinous material (asterisks) and uninvolved tissue (red line), are designated on the scanned image. The nerve profiles are indicated by green arrows and foci of perineurial invasion by turquoise rectangles. (**B**). Enlarged area corresponding to green rectangle in (**A**), with tumor-normal interface showing intratumoral and extratumoral nerves. (**C**). Enlarged area corresponding to green-dashed rectangle in (**B**), showing an intratumoral nerve uninvolved by tumor. (**D**). Enlarged area corresponding to black rectangle in (**A**), with tumor-normal interface showing mostly extratumoral nerves. (**E**). Enlarged area corresponding to black-dashed rectangle in (**D**), showing perineurial invasion. Nerve density is determined per square millimeter by dividing the number of nerve profiles by the respective area of the tumor and normal tissue.

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
