# Peer review of "Nerve Density and Neuronal Biomarkers in Cancer"

_cancers, 2022, doi:10.3390/cancers14194817_

Round 1

Reviewer 1 Report

The paper “Nerve Density and Neuronal Biomarkers in Cancer” by Ali et al, is an important and interesting topic. However, it is an expansive topic and difficult to address meaningfully in a single review article. Consequently, this has led to a diffuse review, the true value of which is difficult to grasp. It would be very valuable to focus on a single group of tumors. By discussing at various points, benign and malignant tumors of the central and peripheral nervous system, the common thread is missing. Some other concerns are mentioned below.  

The authors need to revisit the following points and clarify the information provided.

1)      In myelinated axons, a single Schwann cell encircles only a segment of the axon. The unmyelinated gaps on the axon are the nodes of Ranvier.

2)      Myelin is not the fusion of the cell membranes of Schwann cells and axons. Neither of the cited references supports this statement.

3)      In the paragraph on the perineurium, is it perineural cells or perineurial cells?

4)      Perineural invasion is an example of neurotropism but the two terms are not interchangeable.

5)      Several of the references appear to be incorrect making it difficult to evaluate the statements made in this review. For example, reference 25 is not about melanoma and perineural invasion but rather a commentary on the new AJCC classification. Also reference 29 does not discuss AJCC and CAP recommendations on perineural invasion.

6)      In the summary, the authors claim that they “demonstrated that specific neuronal markers have an important role in tumorigenesis and can serve as important diagnostic and prognostic factors…” or they “confirmed that increased nerve density is associated with worse prognosis in different cancers…” etc. This is a review manuscript and not an original work of research, so these statements should be changed accordingly.   

7)      If two authors are both corresponding, please provide information for second corresponding author.

Author Response

We would like to thank the reviewer for their time and consideration in reviewing our article and helping us improve our research. We have reviewed the reviewer's comments and have included revisions based on each suggestion accordingly. 

Comment: The paper “Nerve Density and Neuronal Biomarkers in Cancer” by Ali et al, is an important and interesting topic. However, it is an expansive topic and difficult to address meaningfully in a single review article. Consequently, this has led to a diffuse review, the true value of which is difficult to grasp. It would be very valuable to focus on a single group of tumors. By discussing at various points, benign and malignant tumors of the central and peripheral nervous system, the common thread is missing

Response: We thank the reviewer for this comment. In accordance with the reviewer's suggestions, we have focused our review on peripheral innervation of extracranial tumors and exclude tumors arising from the central nervous system. This allows us to fully assess the impact of neuronal biomarkers and nerve density in extracranial tumors and allows readers to gain valuable information from a focused review. We have included this information in the Neural Biomarkers and Their Role in Cancer section as follows: “In this review, we will discuss the histology of peripheral nerves, focus on specific biomarkers located in extracranial tumors, and summarize clinical correlates between tumor neurobiology and patient prognosis.” Line 46-48 Page 2

Comment  In myelinated axons, a single Schwann cell encircles only a segment of the axon. The unmyelinated gaps on the axon are the nodes of Ranvier.

Response: We thank the reviewer for this important comment. In accordance we have revised the following sentence, "Unmyelinated gaps on the axon, known as Nodes of Ranvier, are essential for saltatory (discontinuous) and very rapid conduction of depolarization through the nerve fiber." Line 83-85 Page 3

Comment: Myelin is not the fusion of the cell membranes of Schwann cells and axons. Neither of the cited references supports this statement.

Response : We thank the reviewer for bringing this to our attention. As suggested, we revised this section to reflect that myelin sheaths are encapsulating the axons rather that fusing to the cell membrane, as follows: “Each Schwann cell produces a myelin sheath encapsulating a single segment of neuronal axon,” in accordance with the statements made in the reference cited.  Line 82-83 Page 3

Comment:  In the paragraph on the perineurium, is it perineural cells or perineurial cells?

Response : We thank the reviewer for this comment. In the peripheral nervous system, the perineurium surrounds nerve fascicles; its cellular component is composed perineurial cells. We have changed the nomenclature accordingly, to allow for uniformity throughout the manuscript.

Comment: Several of the references appear to be incorrect making it difficult to evaluate the statements made in this review. For example, reference 25 is not about melanoma and perineural invasion but rather a commentary on the new AJCC classification. Also reference 29 does not discuss AJCC and CAP recommendations on perineural invasion.

Response: We thank the review for bringing this to our attention and apologize for this error. We have reviewed all references for relevance and updated the reference list accordingly. Specifically previous reference 25 was updated to  (Gershenwald, J.E.; Scolyer, R.A.; Hess, K.R.; Sondak, V.K.; Long, G.V.; Ross, M.I.; Lazar, A.J.; Faries, M.B.; Kirkwood, J.M.; McArthur, G.A.; et al. Melanoma Staging: Evidence-Based Changes in the American Joint Committee on Cancer Eighth Edition Cancer Staging Manual. CA Cancer J. Clin. 2017, 67, 472–492.) We have also removed reference 29 from our review paper.

Comment: In the summary, the authors claim that they “demonstrated that specific neuronal markers have an important role in tumorigenesis and can serve as important diagnostic and prognostic factors…” or they “confirmed that increased nerve density is associated with worse prognosis in different cancers…” etc. This is a review manuscript and not an original work of research, so these statements should be changed accordingly.   

Response : We thank the reviewer for this comment. We agree with the reviewer's suggestion, as suggested, we have revised this statement to reflect previously published data as follows: “This review demonstrates that specific neuronal markers may have an important role in tumorigenesis and may show potential to serve as important diagnostic and prognostic factors for various cancers.” Line 16-18 Page 1 and “Moreover, increased nerve density may be associated with worse prognosis in different cancers, and that oncologic therapeutics that aim to decrease nerve density may offer benefit to patients.” Line 19-20 Page 1 These statements have been changed accordingly to statements that suggest this is a review paper and not original work of research. We also included the following references to support that neuronal markers and nerve density may be associated with worse prognosis in certain tumors. “Amit, M.; Takahashi, H.; Dragomir, M.P.; Lindemann, A.; Gleber-Netto, F.O.; Pickering, C.R.; Anfossi, S.; Osman, A.A.; Cai, Y.; Wang, R.; et al. Loss of p53 Drives Neuron Reprogramming in Head and Neck Cancer. Nature 2020, 578, 449–454. and “Smits, M. MRI Biomarkers in Neuro-Oncology. Nat. Rev. Neurol. 2021, 17, 486–500.” and “Magnon, C.; Hall, S.J.; Lin, J.; Xue, X.; Gerber, L.; Freedland, S.J.; Frenette, P.S. Autonomic Nerve Development Contributes to Prostate Cancer Progression. Science 2013, 341, 1236361.”

Comment: If two authors are both corresponding, please provide information for second corresponding author.

Response: We thank the reviewer for bringing this to our attention. In accordance, we have added the following information for Dr. Priya Nagarajan as the second corresponding author," Pathology, The University of Texas MD Anderson Cancer Center, Houston, Texas, USA" Line 9-10 Page 1

Reviewer 2 Report

Recent studies have shown that tumor biomarkers and increased nerve density are important clinical tools in differentiating tumors as well as developing effective clinical therapeutics. In this review, the authors reviewed the histological components of a nerve, and neuronal biomarkers found in tumor sites, and discussed clinical correlates between tumor neurobiology and patient prognosis. Their work could be of great value for the development of novel drugs for cancer treatment. The well-written manuscript requires no changes or editing.

Author Response

We would like to thank the reviewer for their thoughtful comments and well wishes. We hope that our manuscript can serve as a important source of information for future researchers.

Reviewer 3 Report

This review mainly demonstrated tumor biomarkers and increased nerve density are arising as important clinical tools in differentiating tumors as well as developing effective clinical therapeutics. They discussed the use of various neuronal biomarkers in cancer, and assessed the impact of increased nerve density on tumorigenesis by detailing the histology of peripheral nerves. It has certain clinical significance. But I think it is not suitable for publication in this journal due to some problems containing related ideas and logic.

Author Response

Comment: This review mainly demonstrated tumor biomarkers and increased nerve density are arising as important clinical tools in differentiating tumors as well as developing effective clinical therapeutics. They discussed the use of various neuronal biomarkers in cancer and assessed the impact of increased nerve density on tumorigenesis by detailing the histology of peripheral nerves. It has certain clinical significance. But I think it is not suitable for publication in this journal due to some problems containing related ideas and logic.

Response: We thank the reviewer for this comment. In accordance with the reviewer's comments, we have made major revisions to our paper and sincerely hope that we are suitable for publication in this journal. Most importantly, to enhance the significance and impact this manuscript we have focused our review on the peripheral innervation of extracranial tumors and exclude all tumors arising from the central nervous system. This allows us to fully assess the impact of neuronal biomarkers and nerve density in extracranial tumors and allows readers to gain valuable information from a focused review. In addition, we have made other important revisions to our paper:

We revised our paper to include the following sentence, “Unmyelinated gaps on the axon, known as Nodes of Ranvier, is essential for saltatory (discontinuous) and very rapid conduction of depolarization through the nerve fiber." This sentence improves semantics to allow readers to better understand Nodes of Ranvier are unmyelinated portions of an axon. Line 83-85 Page 3

In addition, we revised the following sentence to "Each Schwann cell produces a myelin sheath encapsulating a single segment of neuronal axon" This sentence better shows that Schwann cells play a role in the myelination of axons. Line 82-83 Page 3

In addition, we have changed all statements to perineurial cells from both perineurial cells and perineural cells to allow for uniformity throughout the review. 

We have also changed our summary to, “This review demonstrates that specific neuronal markers may have an important role in tumorigenesis and may show potential to serve as important diagnostic and prognostic factors for various cancers.” Line 16-18 Page 1 and “Moreover, increased nerve density may be associated with worse prognosis in different cancers, and that oncologic therapeutics that aim to decrease nerve density may offer benefit to patients.” Line 19-20 Page 1. This allows us to share with the readers that this review may show potential benefit of neuronal biomarkers and increased nerve density on tumorigenesis. It also serves as a purpose to encourage future development of clinical studies that can assess the impact of neuronal biomarkers and nerve density in peripheral tumors.

We hope these major revisions are accepted by the reviewer and that we gain acceptance into this prestigious journal.

We would like to thank the reviewer for their time and consideration.

Round 2

Reviewer 1 Report

Figure 1: Please check labeling for epineurium. What criteria were used for this determination?

Author Response

Point: Figure 1: Please check labeling for epineurium. What criteria were used for this determination?

Response: We would like to thank the reviewer for this important recommendation. We have reviewed our labeling for Figure 1 and have confirmed its accuracy. We have added the histologic criteria, detailed on page 4, lines 106-112. “Epineurium is composed of connective tissue and binds several nerve fascicles together, sometimes along with small blood vessels and lymphatics, to form a neurovascular bundle. It is typically composed of collagen and a few elastin fibers.In larger nerves, the blood vessels feeding nerves, i.e., the vasa nervosum, run longitudinally within the epineurium. The epineurium is thicker proximally; however, in the distal aspect, where only single nerve fascicles are present, the epineurium may disappear completely.” We thank the reviewer for this important suggestion and appreciate their help in ensuring our manuscript presents the most accurate information.

Reviewer 3 Report

I am very grateful to the author for consulting a lot of information and making a number of modifications. The paper has been reversed based on the suggestions, But the concept, structure and writing need lots of modification. I suggest to reject this review. 

Author Response

Point: I am very grateful to the author for consulting a lot of information and making a number of modifications. The paper has been reversed based on the suggestions, But the concept, structure and writing need lots of modification. I suggest to reject this review. 

Response: We would like to thank the reviewer for offering improvements to the manuscript and are very grateful for their time and efforts. In accordance with the reviewer’s suggestions on clarifying the concept and structure of the manuscript, we have added a new section to our manuscript entitled “Mechanism of Interaction Between Nerves and Cancer.” This section details the interactions between nerve and tumor cells which include both direct and indirect interactions. Thank you again for your constructive suggestions that have aided us in building a more succinct, cohesive, and informative paper. We hope these major revisions to our manuscript will allow us to gain acceptance into this prestigious journal.